# Effect of Upstream Valve Opening Process on Dynamic Spray Atomization of Bipropellant Thruster Injector

**DOI:** 10.3390/mi13040527

**Published:** 2022-03-27

**Authors:** Zhen Zhang, Yusong Yu, Jie Cao

**Affiliations:** 1Hydrogen Energy and Space Propulsion Laboratory (HESPL), School of Mechanical, Electronic and Control Engineering, Beijing Jiaotong University, Beijing 100044, China; 21114129@bjtu.edu.cn; 2China North Engine Research Institute (Tianjin), Tianjin 300400, China; jackcao99@163.com

**Keywords:** valve opening process, swirl injector, dynamic spray atomization, bipropellant thruster

## Abstract

In order to develop a new generation of intelligent satellites, fast-response bipropellant thrusters are required to work in minimum impulse mode without limitation. When a valve is opening, the fluctuation affects downstream spray atomization at the injector, which determines the thruster’s impulse performance, involving combustion efficiency and impulse repeatability. Accordingly, the spray atomization under impulse working condition was investigated to optimize the thruster’s dynamic response. The effects of propellant property, switch speed, valve stroke, and throttle orifice layout are respectively compared in simulation cases using OpenFOAM. The fluctuating flowrate caused by valve opening was simulated and then used as boundary conditions for downstream spray. Among these factors, orifice layout plays the most significant roles in transient spray development. Compared with MMH spray, NTO spray from outer swirl injector is more sensitive to upstream fluctuation. When the upstream flowrate stabilizes faster, the atomization stability can also be enhanced, thereby improving the impulse repeatability of thrusters in combustion. This experimental result was in good agreement with the simulation, thereby showing that only when atomization of MMH spray and NTO spray both develop into a steady state within 5 ms after valve opening can the impulse performance be reliably achieved.

## 1. Introduction

The bipropellant thruster is a low-thrust liquid rocket engine used to stabilize or change the attitude of spacecrafts. It operates under various impulse commands with a solenoid valve controlling its switching. The new intelligent satellites require self-adaptive high-precision attitude control; therefore, for thrusters, the minimum impulse command is only a few milliseconds, and the start-up times are up to millions. That is, thrusters are needed to offer minimum impulses with high stability and repeatability in an instant, less than 10 ms. It is beneficial to improve the attitude control efficiency of satellites, prolonging the life span and saving the consumption of propellant in orbit [1].

Injector spray plays a primary role in bipropellant thrusters for spacecraft propulsion systems. The injector is designed to balance the homogeneous mixing of propellant and liquid film cooling by organizing spray atomization. Under the effect of different valve characteristics, whether two propellants can achieve better atomization and mixing in milliseconds is key to improving impulse performance. 

The most direct way to improve impulse performance of the thruster is to use a high-speed solenoid valve and decrease the volume of the injector channel as much as possible. Therefore, we commonly use 3D printing to process the main body of the injector, and then manufacture the inner flow channel by laser processing and chemical etching, in order to minimize the volume of the inner channel.

Traditionally, multiple firing-test runs are needed for validation of the injector design at a high cost. However, spray simulation analysis uses a large amount of data and information, aiming for failure prevention and design optimization. Many researchers have been experimentally and numerically conducting work to study transient swirl spray [2,3,4]. Keller used phase Doppler anemometry and conducted Eulerian–Lagrangian computations to investigate transient spray field phenomena on droplet dynamics and dispersion of an isothermal flow [5]. Snyder used 2D Mie scattering images to test the transient cone angle of pressure swirl sprays from injectors intended for use in gasoline direct injection engines [6]. Baldwin ran a computational study to investigate the influence of transient needle motion on gasoline direct injection (GDI) internal nozzle flow and near-field sprays [7]. Yang analyzed the overall transient spray impingement structure and fuel film formation with different swirl ratios using the CONVERGE CFD code during the intake stroke [8]. Battistoni researched complex off-design fluid dynamics behaviors caused by opening and closing transients that profoundly impact the spray and mixture formation processes, performing highly resolved computational fluid dynamics simulations [9]. 

In the process of transient swirl atomization, fluctuation has a very obvious effect on the spray development. Transient flow in injectors under fluctuation has been studied by some scholars. Cousin et al. conducted simulations to analyze transient flows in high-pressure swirl injectors [10]. The focus was on the transient flowrate during the needle opening, showing the volume of fuel at the start of injection causes a cylindrical jet with a negative effect on fine spray. Khal et al. generated the pressure fluctuation from 5 to 300 Hz upstream of the swirl injector to study the field of injector dynamics [11]. They tested the transient flowrate at the nozzle by using laser diagnostic techniques and the electric conductance method. Evers conducted an experiment with adjustment of ambient chamber pressure fluctuation to research its influence on swirl spray [12]. It was concluded that increasing the ambient pressure increased the size of droplets in all of the spray regions, with decreased secondary atomization. Zhang et al. researched the overall process of double swirl atomization using the methods of large eddy simulation and volume of fluid [13]. In the simulation, they observed two wave patterns in the development of spray, with variable droplet size distribution.

In previous research, the influence of opening characteristics of the solenoid valve on flow fluctuation and transient spray has not been studied. We firstly discuss the effect of high-speed valve opening parameters on dynamic spray, including switch speed, valve stroke, and different orifice layouts. We conduct numerical simulations employing the VOF method to simulate the flowrate fluctuation from the valve outlet and compare these results in different cases. Then, the flowrate fluctuation with swirling effect was taken as a new inlet boundary of a pair of swirl coaxial injectors to analyze the atomization trends of MMH (monomethyl hydrazine) and NTO (nitrogen tetroxide) propellants in milliseconds. 

In the valve switch response test, the open time of the high-speed valve is around 3 ms. This indicates that it is possible for thrusters to work at any impulse command of more than 3 ms. However, whether steady combustion can be achieved depends on the spray development after the valve opening. In the process of the valve opening, the flow is inevitably unstable, and the throttle orifice plate, which is set in the flow passage to control equivalence ratio, generates cavitation, thus causing fluctuation. The transient spray from the swirl injector downstream is significantly determined by the fluctuating flow. The dynamic mesh and the interFOAM solver are used to analyze this fluctuation in the following 5 ms after valve opening under different conditions.

## 2. Methods

### 2.1. Simulation Solver

The inner flow in valve opening and the spray atomization from the swirl injector downstream are simulated using the VOF (volume of fluid) method. Deshpande et al. published a detailed description of the interFoam code in OpenFOAM and showed its capabilities in interface capturing [14,15]. For a low-thrust bipropellant thruster, the diameter of the injector is small, with a low-velocity spray. That is, the Reynolds number is under 2000, and there is no turbulent model used in the following simulations.

Simulations are conducted in two sections. The first is to investigate the fluctuating flow rate in the valve opening process using dynamic mesh. Then, the fluctuating flowrate in these cases can be provided as an inlet boundary condition for the swirl injector spray in the second section. For 2D simulation, the swirling effect can be provided in swirlFlowRateInletVelocity condition, with the RPM (revolutions per minute) value matching its transient flowrate.

### 2.2. Physical Model, Boundary Condition, and Fluid Properties Assumptions

In this simulation study, the focus is on comparisons of primary breakup and spray development. Therefore, 2D simulation cases are conducted to describe the macroscopic shape and stabilization process of the entire spray field, comparing spray visualization and dimensionless total surface area ratio in different cases. The wedge mesh has been proven to be capable of expressing radial dimensional properties [16], as shown in Figure 1.

There are two kinds of axial-symmetric domain used for valve-opening flow at upstream and injector-spray simulations at downstream, respectively, as shown in Figure 2 and Figure 3. In these cases, the wedge mesh with 5 degree is used to simulate inner flow in valve and swirl spray from injectors. The mesh in spray domain is refined as small as 10 μm × 10 μm. It has proven adequate to capture the breakup droplets with an average diameter of above 50 μm, measured by PDPA (phase doppler particle analyzer).

All conditions are listed in Table 1, with different propellants, switch speeds, valve strokes, and throttle orifice layouts.

### 2.3. Validation

In order to validate the simulation method and physics model, the current method was compared with the experimental spray image in the steady-state spray. Due to the toxicity of MMH and the corrosive nature of NTO, deionized water was used to evaluate spray instead. Meanwhile, the simulated spray was also conducted using the liquid property of water for this comparison study. Both of these two inlet conditions are constant flowrate inputs, showing a good agreement in the conical liquid film of umbrella-shape with similar spray angle, breakup length, and droplet size, as shown in Figure 4.

For spray angle, α_tes_ is 69°, and α_sim_ is 70°. For breakup length, *L*_tes_ is 1.5 mm, and *L*_sim_ is also around 1.5 mm. For medium droplet size, the experimental result is 76. μm, measured by PDPA, and the simulated result is around 79 μm. These three parameters can reflect spray characteristics. Spray angle determines the spray shape, breakup length determines the primary breakup, and droplet size determines total surface area of atomization. Therefore, the comparison indicates that the simulation setup is suitable for atomization analysis using the VOF methodology.

## 3. Results and Discussion

The effects of different conditions on transient spray breakup are compared and discussed in this section. The investigated effects are switch speed, valve stroke, and throttle orifice layout for propellants of MMH and NTO, respectively. Under the impulse command of 8 ms, the focus is on the spray process in the following 5 ms, considering the 3 ms delay of solenoid valve opening time, as mentioned above.

### 3.1. Effect of Propellant Properties

NTO is an oxidizer with low surface tension and viscosity, compared with MMH as fuel. Therefore, as for centrifugal bipropellant thrusters, NTO is often designed in outer swirls with more loss of wall friction due to its more easily atomized properties.

Figure 5 shows differences in flowrate fluctuation between MMH and NTO. In the switching process, the throttle orifice plate inevitably causes cavitation in the flow. That is the reason why fluctuation in NTO flow with lower surface tension is more significant. After flowing out from the valve outlet, these two propellants will enter two different separate nozzles. NTO flows into the outer swirls and sprays out from an annular nozzle. MMH flows into the inner swirls and sprays out from a circular nozzle.

At the beginning of injection, the swirl jet forms a conical liquid film under the effect of centrifugal force after ejecting from the nozzle. The front of the liquid film interacts with ambient shearing gas, so that the conical film forms a thin umbrella-shaped structure. A K-H instability wave is generated on the surface of the liquid film due to ambient gas disturbance and the boundary layer effect. What needs to be emphasized is that there is more obvious surface fluctuation just from the nozzle exit than the spray of constant flow. When the surface wave develops to a certain extent with higher amplitude over the film thickness, the liquid film begins to break up into ligaments and droplets. After the first atomization, the liquid ligaments and droplets still move with a high velocity, and continue to split under the aerodynamic interaction, resulting in secondary atomization [17]. The typical development of swirl spray under the condition of inlet fluctuation is shown in Figure 6.

In the spray process, MMH flows in the inner injector can first fill the inside with many cavities under fluctuation [18], and then it gradually develops from unstable to stable with a hollow vortex core after 2 ms. Because the center is occupied by the inner nozzle, NTO liquid film is directly formed and ejected without hollow vortex core, after the filling process with cavitation. The spray angle of the NTO spray is bigger, and secondary breakup, splash, and rebound can be formed after impingement on the chamber wall.

### 3.2. Effect of Switch Speed

Different switch speeds of the armature affect the fluctuation from the valve outlet. As switch speed decreases, MMH flow tends to become more unstable, as shown in Figure 7. While switch speed is 0.25 m/s, the fluctuation amplitude is only strong within the first 2 ms. When the switch speed drops down to 0.125 m/s, the fluctuation amplitude of the MMH flow surges significantly. In contrast, the NTO flow is more sensitive to the effect of armature speed in the opening process. When switch speed is reduced to 0.25 m/s, it is already sufficient to reach the threshold that causes violent fluctuation.

Figure 8 shows visual differences between spray flows at different switch speeds from MMH inner injector and NTO outer injector at 5 ms after valve opening. For MMH spray, when the switch speed is low, violent fluctuation can strengthen the disturbance, so as to obtain comparatively smaller droplets, especially during the first 2–3 ms. However, its stability is worse throughout the whole 5 ms than that of the high-speed switch. Limited by the structural disadvantage of the outer nozzle, NTO spray atomization is relatively more difficult, even though the NTO property is conducive to atomization. Therefore, the atomization of NTO takes a relatively long time. In the instant time of just 5 ms, the high-speed switch contributes to the intact and stable atomization, as shown in Figure 9. However, when it works at a low switch speed, the strong upstream fluctuation can only compensate the breakup of liquid film and ligament at 2–3 ms. When the time reaches 5 ms, the spray is better at high-speed switch with low fluctuation. The reason why the strong fluctuation cannot decrease droplets size significantly is that the damping characteristics of the swirl nozzle structure suppress the influence of upstream fluctuation [19].

### 3.3. Effect of Valve Stroke

The valve stroke determines its opening gap, influencing cavitation generation. In Figure 10a,b, for both MMH flow and NTO flow, increasing the stroke can greatly strengthen the flow stability with lower fluctuation. Even for NTO flow cases of low-speed switch with very violent fluctuation, the fluctuation can also achieve rapid attenuation and damping after increasing the stroke.

Figure 11 and Figure 12 demonstrate the visual differences between spray development at different strokes. In these comparative cases, the liquid film can form an umbrella-shaped structure with a similar hollow vortex core, film breakup length, and spray cone angle. The atomized droplets size and distribution of MMH spray and NTO spray are similar, at both 0.15 mm and 0.25 mm valve stroke. That means decreasing the valve stroke can effectively reduce the oscillation and shorten the time of atomization stability without negatively affecting the atomization quality.

### 3.4. Effect of Throttle Layouts

When the throttle orifice plate is placed upstream of the valve, the instantaneous flow characteristics are different from the layout of a downstream throttle. Compared with the downstream throttle, the orifice plate at upstream leads to a more stable state but much lower transient flowrate, as shown in Figure 13. As liquid flows through the orifice, the flow becomes thinner with increasing velocity and dropping pressure. When the flow expands into a larger area at downstream, velocity decreases, and pressure increases. The larger the downstream area, the greater the transient flowrate decrease. When the orifice plate is at upstream, the entire internal flow channel of the valve is too large to fill at downstream. Therefore, the decreased pressure with the complex channel in the valve results in a low transient flow velocity. When the orifice plate is at downstream, there is only a tiny flow channel of the injector at downstream of the orifice plate, so the injection velocity can still be maintained at a high level under the constant inlet pressure without the throttling effect of valve. Therefore, the swirl injection with lower velocity significantly weakens the aerodynamic interaction, which causes the decrease of atomization quality, especially for NTO spray with the structural deficiency shown in Figure 14. Under the condition of throttle at upstream, the spray angle gets smaller, droplets get bigger, and breakup length gets longer, showing negative effects on the spray and the following combustion. Therefore, throttle orifice layout at upstream is not recommended for fast-response bipropellant thrusters.

### 3.5. Spray Development over Time

In order to investigate the effect of the above factors quantitatively, the total surface area of liquid flow is obtained by integrating all areas of the spray jet interface. Then, the total surface area ratio in each case can be used for comparison by dividing the total surface area of constant flow. When the total surface area ratio stays around 1, that means the injector spray already reaches a steady state. In Figure 15a, for MMH spray, the larger stroke and throttle orifice layout at downstream are more conducive to transient atomization, but the effect is not significant due to its structure. Low-speed switch seems to slightly strengthen the breakup under strong fluctuation, but the resulting instability and inconsistency are negative for the following combustion. In Figure 15b, as for the case of NTO spray, it is sensitive to the change of different orifice throttle layouts. Using the downstream orifice layout can increase the surface area of spray by 2–3 times. Meanwhile, valve stroke and switch speed also have a certain effect on the spray development, but there is no unidirectional tendency. Regarding the above analysis, 1 m/s switch speed, 0.15 mm valve stroke, and downstream throttle layout should be a better combination for optimization.

Figure 16 reflects the dynamic process of transient atomization in both MMH spray and NTO spray, with the optimized options described above. Due to the initial strong fluctuation of the valve switch, the MMH spray reaches a peak at 2 ms, which is 2 ms ahead of the NTO spray. After that, the MMH spray gradually stabilizes after 3 ms, while the NTO spray requires 5 ms to get back to a surface area ratio around 1.

This result is in good agreement with the firing test data. In the short-impulse test, as shown in Figure 17, we can figure out thruster impulse characteristics at different on time commands. At on time of 4 ms command, the impulse repeatability is not good, with 3δ over 60%, because there is only 1 ms, which is inadequate for stable spray and combustion. When on time rises, 3δ decreases to around 40% at the 6 ms command, and just 28% at the 8 ms command. Increasing the switch speed and reducing the delay time of the valve can be the only way to further improve the dynamic response of bipropellant thrusters.

## 4. Conclusions

The transient spray atomization in milliseconds after valve opening was numerically investigated, motivated by improving the dynamic response of bipropellant thrusters. In particular, parametric studies were conducted to study the effects of valve opening parameters on transient atomization of the injector at downstream. 

As the upstream valve switches to open, flowrate fluctuation immediately begins and then gradually stabilizes over time. When switch speed decreases to a certain extent, the fluctuation is strongly enhanced, with negative influence on the downstream spray instabilities. Comparatively, NTO flows are more likely than MMH flows to produce such violent fluctuation. However, increasing the stroke can significantly reduce the fluctuation and improve the flow stability. A downstream throttle layout of the orifice plate allows for greater transient velocity and benefits efficient atomization.

Both MMH spray and NTO spray can first reach a peak of droplets surface area under the strong flow fluctuation, and then MMH spray needs 3 ms for stabilization, while NTO spray requires 5 ms to be stable. Compared with switch speed and valve stroke, throttle orifice layout plays a more important role in influencing the spray development. After comparison, 1 m/s switch speed, 0.15 mm valve stroke, and downstream throttle layout are shown to be a better combination for optimization. The following firing test data proved the simulation results. When on command time is 4 ms, 6 ms, and 8 ms, the thruster repeatability of 3δ is 60%, 40%, and 28%, respectively, showing that an atomization development time of 5 ms is required to obtain a stable and repeatable impulse for a bipropellant thruster.

## Figures and Tables

**Figure 1 micromachines-13-00527-f001:**
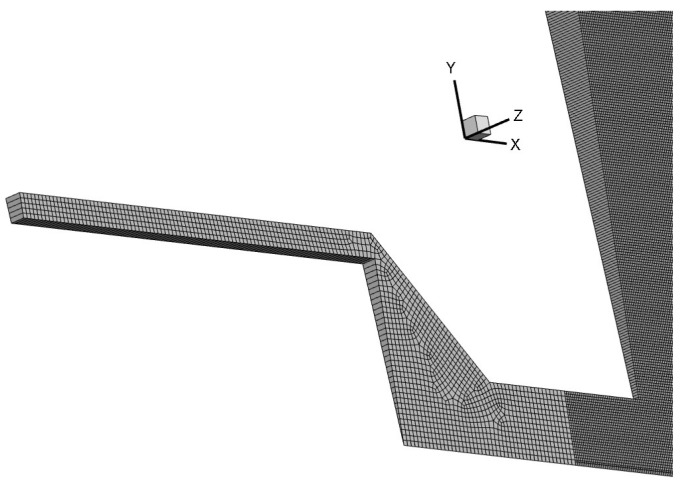
Simulation wedge mesh with 5 degrees to express the flow characteristics.

**Figure 2 micromachines-13-00527-f002:**
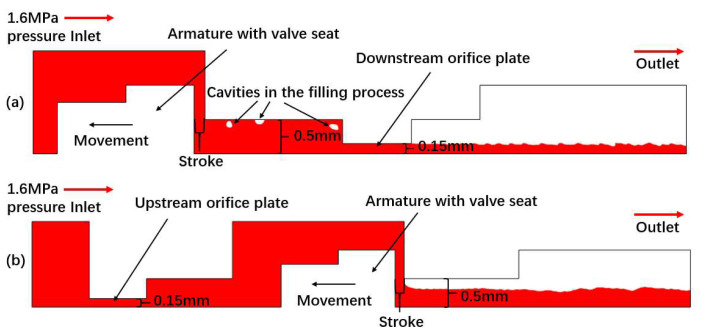
Simulation domains of valve simulation with downstream orifice plate (**a**) and upstream orifice plate (**b**).

**Figure 3 micromachines-13-00527-f003:**
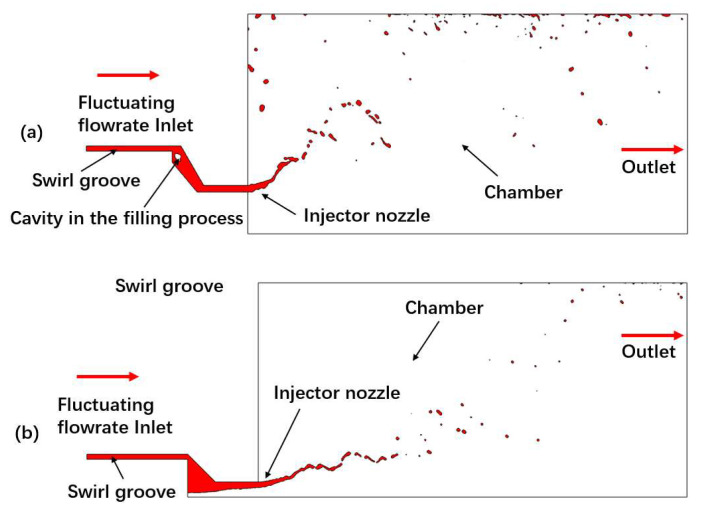
Simulation domains of outer injector spray for NTO and (**a**) and inner injector spray for MMH (**b**).

**Figure 4 micromachines-13-00527-f004:**
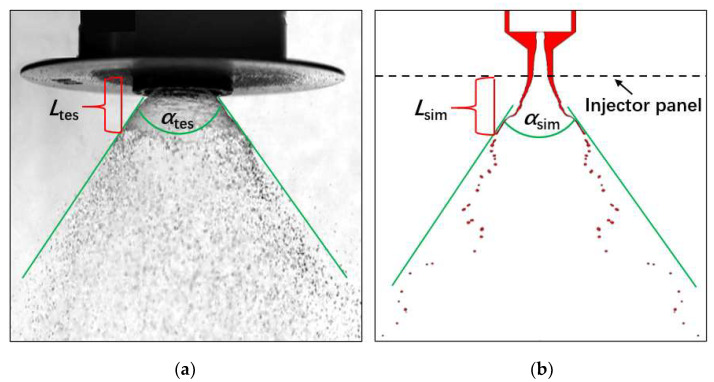
Comparison of validation between experimental image (**a**) and simulated result (**b**) by high-speed camera, with umbrella shape, similar spray angle (α_tes_ and α_sim_), and breakup length (*L*_tes_ and *L*_sim_).

**Figure 5 micromachines-13-00527-f005:**
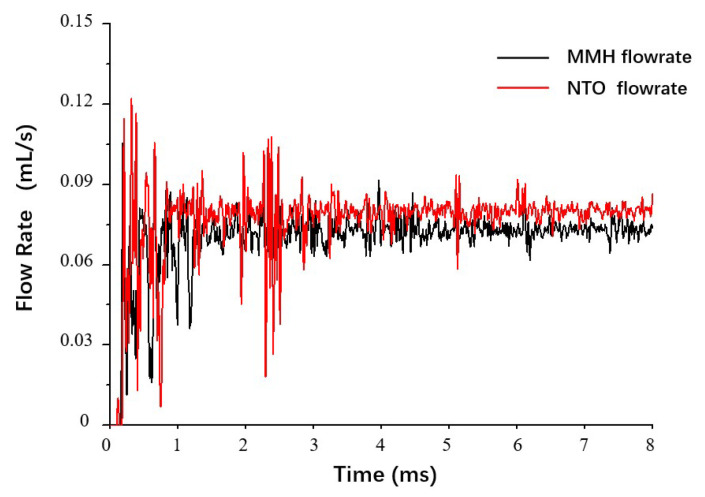
Comparison of flowrate fluctuation between MMH and NTO, with downstream orifice plate at 0.5 m/s switch speed and 0.25 mm stroke.

**Figure 6 micromachines-13-00527-f006:**
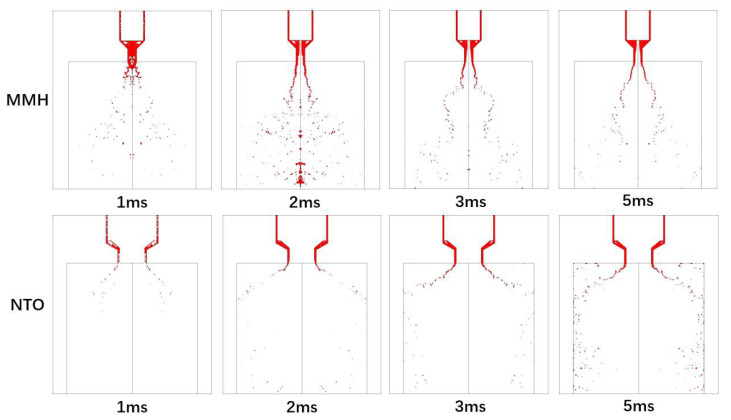
The development of MMH and NTO transient spray atomization over time after valve opening.

**Figure 7 micromachines-13-00527-f007:**
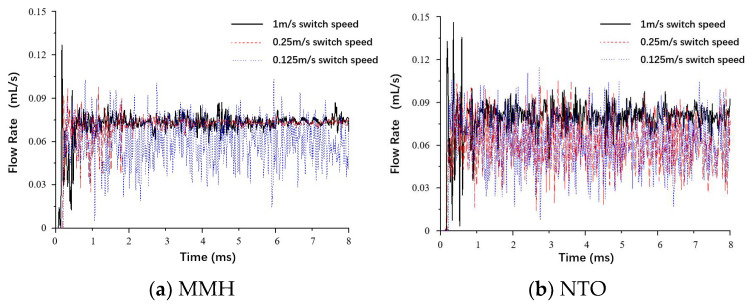
Comparison of MMH (**a**) and NTO (**b**) flowrate fluctuation under different switch speeds, with downstream orifice plate and 0.15 mm stroke.

**Figure 8 micromachines-13-00527-f008:**
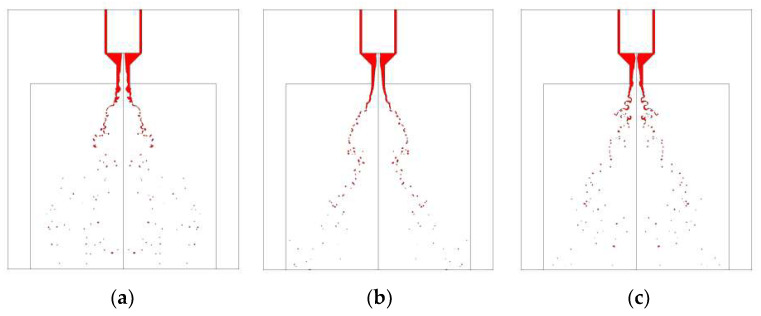
Comparison of MMH spray development at 5 ms under different switch speeds: (**a**) 0.125 m/s, (**b**) 0.25 m/s, (**c**) 1 m/s, with downstream orifice plate and 0.15 mm stroke.

**Figure 9 micromachines-13-00527-f009:**
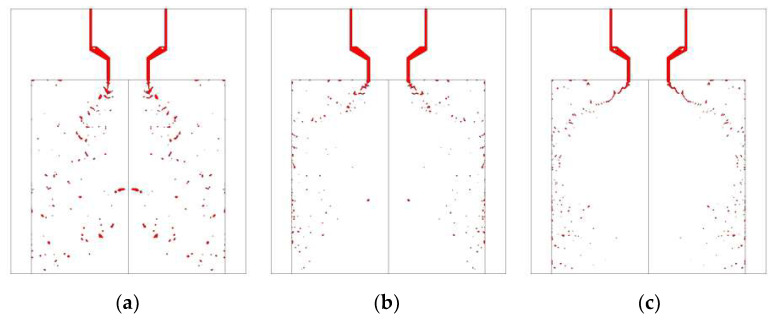
Comparison of NTO spray development at 5 ms under different switch speeds: (**a**) 0.125 m/s, (**b**) 0.25 m/s, (**c**) 1 m/s, with downstream orifice plate and 0.15 mm stroke.

**Figure 10 micromachines-13-00527-f010:**
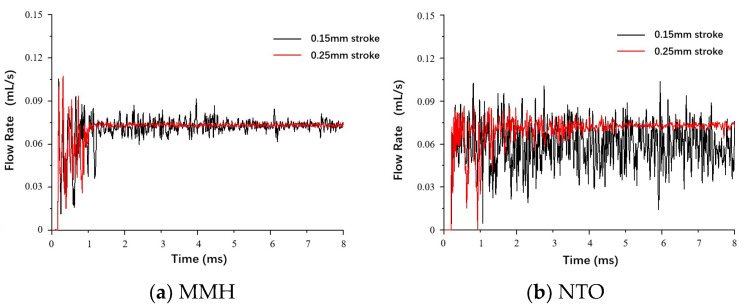
Comparison of MMH (**a**) and NTO (**b**) flowrate fluctuation under different switch speeds, with downstream orifice plate and 0.15 mm stroke.

**Figure 11 micromachines-13-00527-f011:**
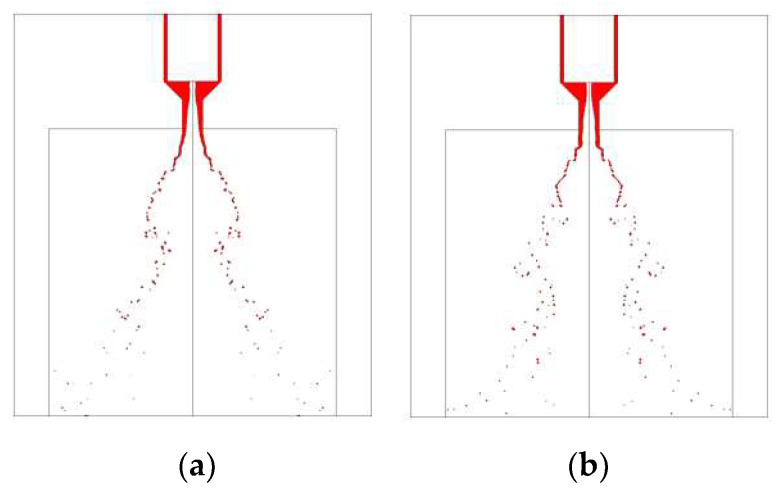
Comparison of MMH spray development at 5 ms under different valve strokes: (**a**) 0.15 mm, (**b**) 0.25 mm, at 0.25 m/s switch speed with downstream orifice plate.

**Figure 12 micromachines-13-00527-f012:**
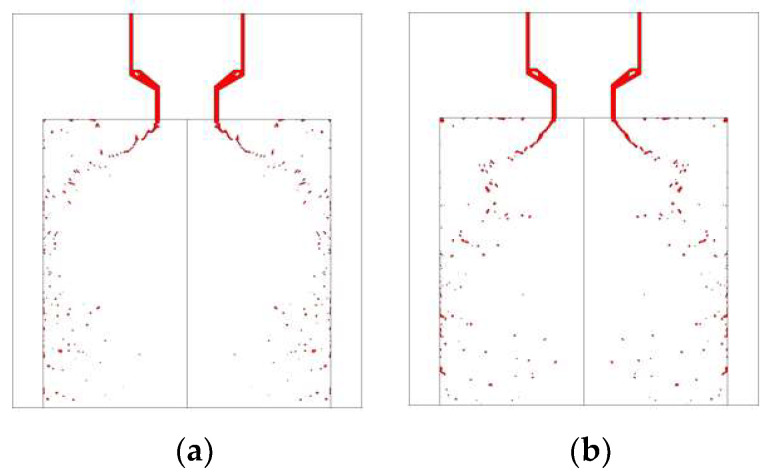
Comparison of NTO spray development at 5 ms under different valve strokes: (**a**) 0.15 mm, (**b**) 0.25 mm, at 1 m/s switch speed with downstream orifice plate.

**Figure 13 micromachines-13-00527-f013:**
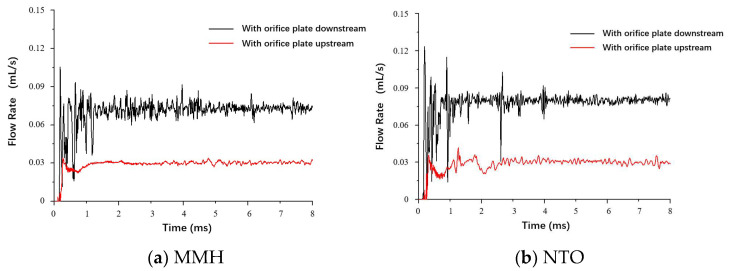
Comparison of MMH (**a**) and NTO (**b**) flowrate fluctuation under different switch speeds, with downstream orifice plate and 0.15 mm stroke.

**Figure 14 micromachines-13-00527-f014:**
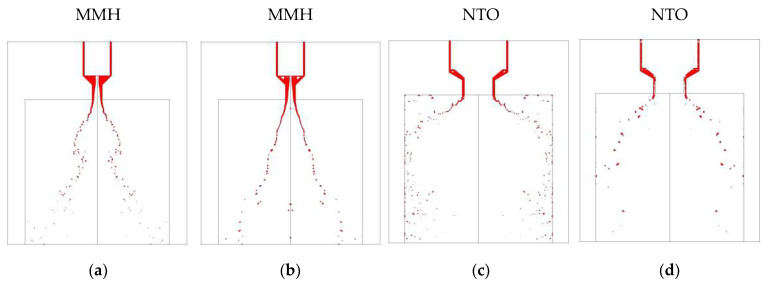
Comparison of MMH spray development at 5 ms under different valve strokes: (**a**) 0.15 mm, (**b**) 0.25 mm, at 0.25 m/s switch speed with downstream orifice plate.

**Figure 15 micromachines-13-00527-f015:**
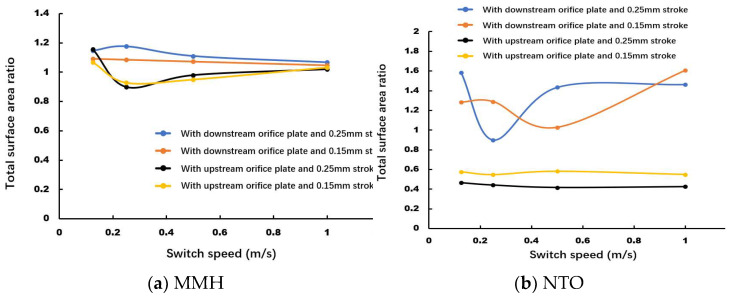
Comparison of MMH (**a**) and NTO (**b**) flowrate fluctuation under different switch speeds, with downstream orifice plate and 0.15 mm stroke.

**Figure 16 micromachines-13-00527-f016:**
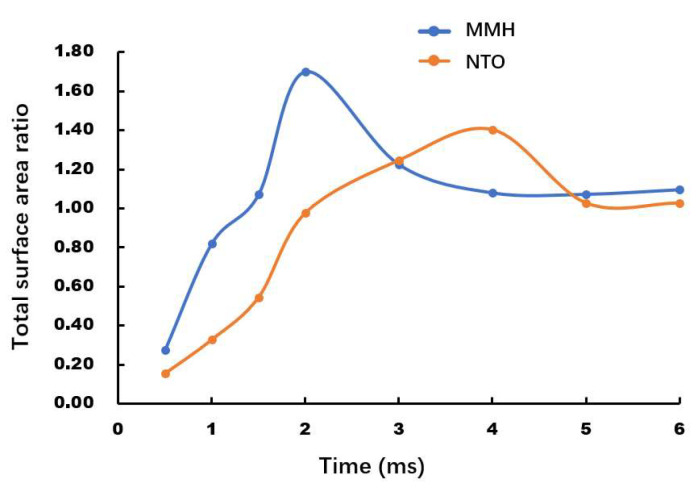
Total surface area ratio over time at 1 m/s switch speed, 0.15 mm stroke, with downstream throttle layout.

**Figure 17 micromachines-13-00527-f017:**
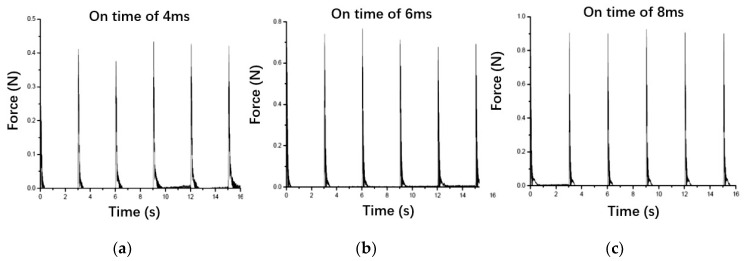
The short-impulse test data of a thruster at different on time commands with 1 m/s switch speed, 0.15 mm stroke, and downstream throttle layout.

**Table 1 micromachines-13-00527-t001:** Conditions of simulation cases.

Propellant Properties	Ambient Pressure (Pa)	Valve Switch Speed (m/s)	Valve Stroke (mm)	Orifice Throttle Layout
MMH, NTO	100	0.125, 0.25, 0.5, 1	0.15, 0.25	Upstream, downstream

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
