# Peer review of "Effect of Upstream Valve Opening Process on Dynamic Spray Atomization of Bipropellant Thruster Injector"

_micromachines, 2022, doi:10.3390/mi13040527_

Round 1

Reviewer 1 Report

The present manuscript investigated the effect of upstream valve opening on dynamic spray atomization of bipropellant thruster injector.
The topic is important, however, the reviewer felt that the manuscript needs some issues to be clarified.

1. Scope of this journal, Micromachines, encourages submissions on significant and original works related to all aspects of micro/nano-scaled structures, materials, devices, systems as well as related micro- and nanotechnology from fundamental research to applications. 
The section of Engineering and Technology deals with micro/nano fabrication and manufacturing: deposition, lithography, patterning, etching, surface micromachining, bulk micromachining, laser fabrication, biofabrication, 3D printing, self-assembly, etc., design and optimization principles of micro- and nanosystems; micro- nanosystems and advanced technologies for engineering applications.
This manuscript has been submitted to the Special Issue of Ocean MEMS, however, the topic of the manuscript belongs to the Aerospace field.

The Editors should consider carefully if the submitted manuscript fits with the scopes of the journal and special issue.
Based on the reviewer's knowledge, the reviewer suggests submitting to the aerospace-related journal.

2. In the introduction section, some research trends regarding the transient swirl spray in other fields have been briefly explained.
Some research trends of the transient swirl spray for the rocket injector applications (including MMH-NTO bipropellant thruster) need to be added.

3.  Please explain clearly the originality and differences of the present study when compared with other previous studies of the transient swirl spray for the rocket injector applications (including MMH-NTO bipropellant thruster).

4. The authors need to clarify if the experimental spray image in Figure 4 was obtained with MMH-NTO propellants or not. 
When considering Figure 5, the experimental spray image in Figure 4 seems to consider only one kind of propellant. So it is not clear if the propellant in Figure 4 is MMH, NTO, or other liquid.

5. The authors claim that the numerical and experimental results are similar to each other.
   To clarify this, the authors need to provide and compare the exact values of spray angle, breakup length, and droplet size. 

6. Please add the detailed reason why the downstream throttle layout of the orifice plate allows for greater transient velocity and benefits efficient atomization.

Reviewer 2 Report

The article raises very interesting scientific issues. Research concerns MMH spray and NTO spray. At the introduction and at the end of the summary, the work should be new in relation to the current state of knowledge. The authors present what they did in the article. A reader who is unfamiliar with the subject may not find out if the work is innovative. Therefore, it is worth mentioning these issues in the summary in the introduction. The introduction is well structured. Can you provide publications on the practical application of these solutions? I recommend adding a few items. The method description is correct. The description of the method is brief. The illustrations are correct. The designations of the formulas are correct. The subsection titles are thoughtful and accurate. All photos are in good resolution, the markings are clear and well prepared. The results are well described. Supported by numerical values, figures and charts. The work is very well prepared in terms of editing. Paragraph on line 173. Separate the title of the drawing with a blank line. Similarly lines 212, 246. The entire chapter on resolute results and description is correct. I do not see any factual errors. Summary. I recommend that you list the most important conclusions from dashes and give percentages or numbers.

Round 2

Reviewer 1 Report

All the issues are closed. The manuscript can be accepted for publication.